# Testing Potential Transfer Effects in Heritage and Adult L2 Bilinguals Acquiring a Mini Grammar as an Additional Language: An ERP Approach

**DOI:** 10.3390/brainsci12050669

**Published:** 2022-05-20

**Authors:** Sergio Miguel Pereira Soares, Tanja Kupisch, Jason Rothman

**Affiliations:** 1Department of Linguistics, University of Konstanz, 78464 Konstanz, Germany; sergio-miguel.pereira-soares@mpi.nl (S.M.P.S.); tanja.kupisch@uni-konstanz.de (T.K.); 2Language Development Department, Max Planck Institute for Psycholinguistics, 6525 Nijmegen, The Netherlands; 3Department of Language and Culture, UiT the Arctic University of Norway, 9019 Tromsø, Norway; 4Nebrija Research Center in Cognition, University of Nebrija, 28015 Madrid, Spain

**Keywords:** L3/L*n* acquisition, EEG/ERPs, Mini-Latin, morphological case, adjective position

## Abstract

Models on L3/L*n* acquisition differ with respect to how they envisage degree (holistic vs. selective transfer of the L1, L2 or both) and/or timing (initial stages vs. development) of how the influence of source languages unfolds. This study uses EEG/ERPs to examine these models, bringing together two types of bilinguals: heritage speakers (HSs) (Italian-German, *n* = 15) compared to adult L2 learners (L1 German, L2 English, *n* = 28) learning L3/L*n* Latin. Participants were trained on a selected Latin lexicon over two sessions and, afterward, on two grammatical properties: case (similar between German and Latin) and adjective–noun order (similar between Italian and Latin). Neurophysiological findings show an N200/N400 deflection for the HSs in case morphology and a P600 effect for the German L2 group in adjectival position. None of the current L3/L*n* models predict the observed results, which questions the appropriateness of this methodology. Nevertheless, the results are illustrative of differences in how HSs and L2 learners approach the very initial stages of additional language learning, the implications of which are discussed.

## 1. Introduction

A crucial difference between modeling second and third language acquisition is the number of previously acquired mental grammars/languages the learner (or internal parser) has available for transfer. In other words, when one starts learning a second language (L2), crosslinguistic influence (CLI) (i.e., transfer) can only come from the L1 or not at all [1,2,3,4]. In contrast to L2 acquisition, understanding and capturing transfer in an L3/L*n* environment is more convoluted, given that more factors can affect transfer. These include the number of sources (no source, L1 transfer, L2 transfer, or a combination of both) [5,6,7], the timing in acquisition (ab initio versus over development—perhaps at iterative stages), the differential impact multilingualism might elicit on cognitive development and the context of previous language learning [8]. Therefore, even if the underlying language processes and cognitive mechanisms do not fundamentally differ, it has been posited that L2 and L3/L*n* acquisition need to be understood as two separate fields of (overlapping) study [9,10,11].

### 1.1. L3/Ln Literature: Models and HSs

Various models have been proposed in the L3/L*n* literature to account for transfer patterns. The *Cumulative Enhancement Model* (CEM) [12] predicts morphosyntactic transfer from both source languages (the L1 or the L2) as long as it is facilitative for the property of the target L3. The *L2 Status Factor* (L2SF) [13,14,15,16,17] contends that the L2 has a privileged role as a transfer source. At its core sits the Declarative/Procedural (DP) model [18], which states that grammars of the first and subsequently acquired languages are stored in different cognitive systems (i.e., procedural memory for the L1 and declarative memory for subsequent non-native languages). Following the DP’s theorizing, it is likelier for transfer to take place within languages of the same system (L2 to L3/L*n*) than between different systems (L1 to L3/L*n*).

More recent models argue that typology, i.e., the underlying structural similarity (also referred to as proximity) between source languages and L3/L*n,* plays an important role in source language selection. The Typological Primacy Model [19,20,21,22] allows for either the L1 or the L2 to be the transfer source. However, unlike other models, the TPM stipulates that only one of the two previously acquired grammars is the source of transfer in its entirety (holistic/full transfer) for the initial L3/Ln interlanguage grammar [8]. The parser determines which source language is the most economic grammar to be transferred via linguistic (holistic structural) proximity. Proximity is determined via a hierarchy of linguistic cues, which indicates to the parser the order of transfer source selection [8,21]. The parser starts by examining the lexical level of previously known languages and contrasts them with the newly acquired language. If the lexical level is sufficiently informative to make a decision (transfer from L1 versus L2), then transfer will be initiated at this stage. If there is insufficient cross-over between the source languages and the L3/L*n*, then the next level of cues is phonology/phonetics. If this second level is insufficient to establish similarity, first morphology, then syntactic structure kick in.

Similar to the TPM, the Linguistic Proximity Model (LPM) [23] claims that transfer can happen from the L1 or the L2. However, in contrast with the TPM, transfer is not wholesale but property-by-property. Akin to the TPM, structural proximity is deterministic for transfer to the L3. The LPM does not assume representational copying, but incremental learning as a result of processing, making early L3 unstable representations increasingly stable. Therefore, the LPM is more focused on describing and capturing processes of multilingual language development rather than fixating on the initial stages of L3 interlanguage representation. Similar to the LPM, the Scalpel Model (SM) [24] also claims selective (property-by-property) L3/L*n* transfer, putting forward additional variables that determine L3/L*n* development. The core idea is that multilingual language selection takes place in a *scalpel-like* manner: properties are carefully selected for transfer by the scalpel either from the L1 or the L2 when more facilitative. However, the scalpel can make mistakes, leading to non-facilitative transfer. 

In sum, determining whether L3/L*n* transfer is *holistic* (from either L1 or L2) or *selective* (property-by-property from both L1 and L2) and the timing, i.e., at the *initial stages* or staggered over the course of L3/L*n development*, underscores the main discussions in the current L3/L*n* literature [8,25,26,27]. 

In the growing field of L3/L*n* morphosyntax acquisition, little work has been carried out with heritage speakers (HSs) (see [28,29] for an updated review of available studies), and few have compared HSs with L2 learners under the same conditions of L3/L*n* exposure. To date, most of the literature has been concerned with L3 learners whose L3 represents a second sequential foreign language. Yet, HSs are special in having two naturalistically acquired grammars/languages to draw on: the minority language acquired mostly at home and the societal majority language. Therefore, studying HSs might help one to understand the inherent diversity between and within HSs and late L2 learners while shedding unique light on the applicability of existing L3/L*n* models in multilingual language processing [8,30]. The present study aimed at specifically testing models hypothesizing typological/structural proximity as a major factor in initial language selection for L3/L*n* transfer. That is, either via full transfer of one or the other language system as advocated by the TPM [19,20,21,22], or selectively, i.e., entertaining property-by-property transfer following the parser’s assessment of each domain of grammar (LPM and the SM) [23,24]. 

### 1.2. Online Brain Methods and Mini Grammars 

There are compelling reasons to use complementary online methodologies in discussions of L3/L*n* morphosyntax transfer. Neuroimaging methods, such as EEG, offer a window into language processing in real-time and have been successfully used for addressing similar questions in L2 acquisition [8,31]. EEG/ERPs capture involuntary brain responses at the brain level with optimal temporal resolution. Thus, they are not directly influenced by metalinguistic variables the way behavioral methods can be. Additionally, EEG data can reflect the underlying grammatical representations that feed into various language-related brain signatures where ungrammaticality is present [32]. By extension, if L3/L*n* transfer selection is a copy of underlying mental representations of the L1 or L2 at the very early stages, as advocated by the TPM [22], then one might expect evidence of neural signatures to be able to support such a claim or question it. 

The use of a natural (or artificial) mini grammar paradigm complements the use of EEG/ERPs. In a mini language paradigm, only a (minimal controlled) subset of a naturalistic language is taught (restricted vocabulary and two grammatical rules in our particular case), providing control for specific research questions to be addressed. In the present study, we control the input of the target L3/L*n* across participants in all relevant aspects: quantity and quality, in the same context and manner of exposure and at the same development timing, i.e., at a true ab initio stage of L3/L*n*. Doing so evokes increased confidence that the limited exposure, controlled as it is in a mini language paradigm, is sufficient to promote transfer selection. 

To date, there are few studies that have used EEG in the landscape of L3/L*n* interlanguage transfer and acquisition. Only one recent study has combined EEG with mini grammar initial exposure [33]. Native Spanish L2 English speakers were tested on first exposure with two artificial languages based on either the English or the Spanish lexicon. Both had the same novel morphology for gender and number on nouns and corresponding agreement for determiners and adjectives. This made the mini grammars similar to Spanish irrespective of their lexical base. Participants were split into two groups and received implicit training either in Mini-English or Mini-Spanish. Findings showed that gender violations produced a highly localized fronto-lateral negativity in the group exposed to Mini-English in an early time window (200–500 ms). The group trained on Mini-Spanish elicited a more broadly distributed positivity in the 300–600 ms time window. Although the authors’ initial prediction was to find markers of syntactic violation ((N400)-P600) for the Spanish mini grammar group, the most salient finding was a P300-like effect that appeared only for the group which was exposed to Mini-Spanish. The P300 is not a common signature found in (un)grammaticality processing per se, but it is well attested for information processing, context updating and attention literature [34,35,36,37]. The authors interpreted the P300 as a sign of increased allocation of attentional resources (happening before grammatical transfer takes place at the level of representation), likely driven by cues in the target language. In other words, despite both groups being matched for nativeness in Spanish and proficiency in L2 English, it was the target language that made the difference. If exposed to a mini grammar based on the Spanish lexicon, one could focus attention differentially than if exposed to the English-based one even though each participant has comparable experience with the previous languages. Given the specific EEG signatures, these results reflect processes happening before transfer source selection in L3/L*n* takes place. This is potentially the precursor stage of when the parser makes the decision of which language to go with for transfer. The authors questioned whether the exposure time was insufficient to capture signatures of grammatical transfer, although the P300 might indicate something that would later lead to it (and differentially between the two groups). They suggested that longer sessions or a consolidation phase (time between first exposure and re-introduction of the mini grammar before EEG testing) might lead to greater chances of capturing signatures of (differential) grammar processing. In the present study, this methodological approach is adopted—more exposure and a consolidation period—to address similar research questions. 

### 1.3. Case and Adjective Placement in Italian, German and Latin

The present study investigated case morphology and the position of attributive adjectives. Case morphology in Latin is similar to that of German at the level of underlying syntactic features, yet it differs in morphological realization. In German, case is mostly marked on articles and adjectives [38], and in Latin, on adjectives and nouns. By contrast, the relative position of nouns and adjectives is similar between Italian and Latin, where adjectives in NPs appear mostly postnominally, while they precede the noun in German. 

***Case and its acquisition***. Latin has an extremely rich inflectional system. Nouns are inflected for number and case, whereas adjectives are inflected for number, case and gender [39,40,41,42]. Nouns can have masculine, feminine or neuter gender, and there are six cases. Herein, we focus on nominative and accusative cases. German has four cases, including nominative (marking subjects) and accusative (marking direct objects), which are also found in Latin. German also distinguishes three genders and number. Italian has evolved a reduced inflectional system in comparison to Latin [43], although remainders of the original case system can still be seen in the pronoun system, where the difference between subject and object pronouns is marked (e.g., *Io*-_NOM_ “I”, *me*-_ACC_ “me”). Examples (1) and (2) compare case marking (for masculine/feminine and nominative/accusative only) between the three languages, as relevant to the present study: 
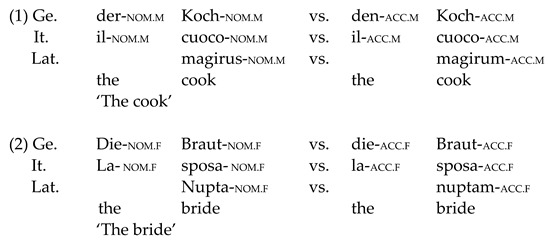


Given the mini artificial Latin paradigm tested here, an important question is how complex case morphology is acquired by early bilingual speakers. Although case is generally considered to be vulnerable in HSs and L2 learners [44,45,46,47], this does not seem to hold for the majority language of the HSs (and German is the majority language in the present study).

***Adjective placement and its acquisition***. Adjective placement in relation to the noun is similar between Italian and Latin. In these languages, adjectives can be placed before and after the noun, although the latter is the default position (see [48,49,50,51,52,53,54]). In both languages, different factors influence the position of the adjective [54,55]. Changes in noun–adjective order (and word order more broadly) in Latin have been reported to vary diachronically [55] and depend on the style of the author [56,57,58,59] (since the participants in the present experiment have no previous knowledge of Latin (rules), Mini-Latin adjectival position has been created to follow Italian word order rules [52,53] and, thus, to be the linguistic domain that differs from case morphology when it comes to adjudicating between models of L3/L*n* transfer and acquisition). Similar to Latin, the position of adjectives in Italian is variable, depending on semantic factors [52,53]. Adjectives referring to nationalities and colors, for example, always follow the noun [52,60]. For the present study, only adjectives of nationality have been selected since they appear postnominally in Italian. Since nationality adjectives do not exist in Latin, they were created following Latin phonotactic and lexical rules (e.g., *tidonus*, “American”) (Appendix A). Opposingly, in German, adjectives virtually always precede the noun [61]. The contrast for adjectival position between the three languages investigated in the present experiment is illustrated in (3): 
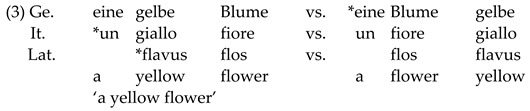


In order to successfully test the validity of the predictions of the L3/L*n* models in this Mini-Latin paradigm, a crucial precondition needs to be met, i.e., that the HSs are in fact able to transfer Italian adjective–noun order into Latin. In other words, their grammatical order of nouns and adjectives needs to follow the Italian grammatical rules reviewed above [52,60], rather than being subject to transfer from German. Previous work on the bilingual acquisition of Italian shows that adjective placement is acquired with a delay [62] and that children overuse the prenominal position [63]. Nevertheless, at adult ages, Italian HSs show no CLI from German in their spontaneous speech [49]. Instead, they seem to overaccept postnominal adjectives., i.e., exaggerating the contrasts between their languages rather than the similarities. This indicates that they are aware of the postnominal order. 

To ensure that the HSs have the rule/grammatical representations at hand (i.e., both case morphology in German and Italian adjective–noun placement), an “Adjective and Case Assignment Task” (see Section 2.5) was administered at the end of the EEG/ERP session (as advocated in a similar experiment with a mini language) [33].

### 1.4. Research Questions and Hypotheses

In light of the review above, the two guiding research questions for the present study are: 

**RQ1.** 
*Can neurophysiological signatures (EEG/ERPs) inform models of multilingual morphosyntactic transfer?*


**RQ2.** 
*What do data from HSs bring to bear for answering RQ1 as well as understanding how they might differ from L2 learners when acquiring additional languages in adulthood?*


ERP predictions (see Table 1) for the relevant models of L3/L*n* transfer were based on sentence processing work in native speakers and adult L2 non-native speakers in combination with theoretical predictions and actual behavioral patterns from previous L3/L*n* work (see [31] for an initial theoretical set of predictions in L3/L*n* and [33] for an L3/L*n* study with ERPs testing these predictions—i.e., (N400)-P600 as markers of syntactic violations/representational transfer). Three decades of literature on native speakers have produced strong evidence for an association between (morpho)syntactic sentence violations and the P600 ERP component [64,65,66]. However, there is no one-to-one correspondence. That is, the P600 is not specific to language and therefore is not a direct marker of underlying grammatical representation. Rather, it is a signature of the processes that happen when ungrammaticality is encountered, reflecting repair or reanalysis, that is, late integrations and repair of syntactic, semantic and thematic information [67,68]. 

Non-native speakers show more variability in their neurophysiological responses to agreement. They tend to adapt and vary in relation to factors such as L2 proficiency and language combinations [69] and show both N400 and P600 components. Whereas the N400 is mostly found in low proficiency L2 speakers for agreement violations in novel (only present in the L2) and sometimes non-novel (shared in both the L1 and L2) features [70,71], non-novel features (already present in the L1) tend to produce P600 native-like responses, even at low levels of proficiency [72,73,74]. The P600 emerges for novel features at high proficiency levels too [69,75]. Crucially, the fact that exposure and experience with grammatical features lead to native-like electrophysiological indices at early stages suggests (for the purposes of this study) that linguistic transfer can be captured and disentangled from other potential sources of target-like behavior in feature agreement at the earliest stages of non-native language development.

Whereas a lot of work has been carried out on (morpho)syntactic (and semantic) agreement violations, word order with electrophysiology, in particular adjective–noun order, is understudied. Courteau et al. investigated the processing of adjective–noun order violation in French by means of ERPs [76]. Similar to Italian, French adjective word order is freer than in German and English, but not completely free. Specifically, the position of adjectives relative to nouns can be predicted by their specific lexico-semantic category [77]. This means, for example, that adjectives of size/magnitude (“big”) occur prenominally, and color/nationalities postnominally (see Section 1.3). The authors set out to investigate whether, similar to English [78,79,80], adjective–noun order violations elicit a biphasic N400-P600 effect and if there are fundamental differences in the way the two typologies of adjectives are processed. Their findings partially corroborate previous results in English, such that incorrect adjective order elicits first an N400 (reflecting a mismatch at the semantic category level) followed by a P600 (reflecting a typical process of sentence reanalysis). However, this was only the case for prenominal adjective violations. Postnominal ones elicited only a P600. Thus, prenominal and postnominal adjectives seem to be processed slightly differently, likely to reflect different expectancies about upcoming linguistic structures [81]. 

Based on the above ERP literature, predictions were formulated as reported in Table 1. The TPM claims transfer to be holistic, i.e., only one of the previous grammars (either L1 or L2) is copied as a whole for the first L3 interlanguage. Given the TPM’s claimed hierarchy of cues for the parser to make (in [25] terms, the “big decision” for which previous grammar should serve as the feature specifications of the initial L3/L*n* interlanguage grammar), the HSs should choose Italian, and the German L2 group should choose English. The ERP predictions are presented accordingly. On the other hand, both the LPM and the SM predict selective transfer based on structural similarity at each property level. As a result, they seem to predict that for the HSs, Italian can be the source for adjectival syntax and German* is potentially possible for case. For the German L2 group, German* should be equally available for the transfer of case and since neither English nor German provide the target syntax, there should be a null effect for adjectival word order. 

## 2. Materials and Methods

### 2.1. Participants

The research procedures in this study were approved by the University of Konstanz Research Ethics Committee. Before taking part in the experiment, participants gave written informed consent and confirmed no contraindication to the EEG investigation. Participants who presented a neurological condition were excluded from this study. Furthermore, participants were compensated for their time. Two groups of participants were recruited, 15 HSs of Italian living in Germany (females *n* = 12) and a comparison group of 28 L2 learners (females *n* = 19) (a total of 48 participants were recruited. Five dropped out for various reasons (see 2.4.1), resulting in a final sample of 43 participants). The L2 learners spoke German as their L1 and English as an L2, whereas the HSs had Italian as their L1 and either acquired German simultaneously as their second L1 (2L1) or sequentially before age 4 (and L2 English). Some of the participants acquired a variety of other languages in school/university settings (e.g., French, Spanish, Arabic, Russian, Chinese, Norwegian or Swedish), but their proficiency was low and they did not use those languages actively. All participants were recruited from either the Lake Constance region or North Rhine-Westphalia and tested at the Universities of Konstanz, Düsseldorf and Cologne. The age range was 18–35 y.o. (mean German L2 group = 25.10 y, SD = 3.61 y; mean HSs = 24.27 y, SD = 4.01 y). The age of first exposure to the L2 was 9.46 y (SD = 1.94 y) and to the 2L1 was 2.21 (SD = 1.63 y). 

### 2.2. Background Measures

As a means of assessing language proficiency in Italian and German, versions of the language placement tests originally created by [82] (p. 80), known as the DIALANG test battery, were adapted (available on https://dialangweb.lancaster.ac.uk/ accessed on 29 March 2022). The test consists of 50 real words and 25 pseudo-words. YES and NO responses were required and collected by means of a button box. The versions (Italian and German) used here are the same as in [83]. Items appeared in the center of the screen one at a time. Once the participant’s decision was met (YES—green button and NO—red button) and the answer was recorded, the next word was presented (self-led task programmed in *Presentation*^®^). Prior to the start of the experiment, participants were instructed to press YES if they thought the word on the screen existed and NO if they did not. Since the items were presented locally (and not directly on the DIALANG webpage), the scoring could be carried out in the lab. Following [82] and work from our laboratory [83], scoring consisted of simply the sum of all correct answers (i.e., one point for each correctly identified word or non-word—“Simple Total Score”). On the other hand, the online version of the test is stricter and penalizes the acceptance of non-words. There were no differences in German proficiency between the two groups (mean German L2 group = 94.18%, SD = 5.4; mean HSs = 94.22%, SD = 4.74) when performing a two-tailed T-test (*t*(32) = −0.02, *p* = 0.98). For the HSs, the Italian version resulted in a proficiency score of 75.47% (SD = 10.65). 

English proficiency was documented online via the LexTALE [84]. The test consists of 5 min YES–NO vocabulary judgments similar to the DIALANG and predicts English vocabulary knowledge and (possibly) general English proficiency. The test comprises 40 words and 20 nonwords (for a total of 60 items) and 3 practice trials (presented at the beginning to familiarize the participants with the task). Participants were instructed to press the green button (on a button box) if they thought the presented item was a real word, and the red button if otherwise. Scores were calculated as percentages following the automatically returned online calculation method. There were no differences in English proficiency between the two groups (mean German L2 group = 69.37%, SD = 12.86; mean HSs = 62.42%, SD = 10.86) when performing a two-tailed T-test (*t*(33) = 1.85, *p* = 0.07).

Participants also completed the Language and Social Background Questionnaire (LSBQ) [85], which assesses language use in various social aspects, settings (e.g., in society and at home) and activities. A factor calculator within the questionnaire computes three weighted composite scores (as the sum of different relevant questions): the amount of language use at home, in social contexts and overall language proficiency. Higher scores indicate higher language use and proficiency. In addition to these measures, age of acquisition of the second language (AoA), length of exposure to the second language (LoE) and socio-economic status of the mother (SES) were assessed (see Appendix A).

### 2.3. Latin Mini Language

A Latin mini language (Mini-Latin) based on Latin vocabulary and grammatical rules was created (see [33] for another L3 ERP study using new languages ab initio and [86,87,88,89] for studies using artificial Latin mini languages to investigate various aspects of multilingual language acquisition and processing). Mini-Latin contained 14 nouns (7 masculine and 7 feminine), 8 transitive verbs and 6 adjectives (see Appendix A) either borrowed from other studies investigating Latin or found in online corpora. 

Latin words were selected so that they were not cognates in either Italian or German, in an effort to avoid any learning strategies that could directly induce superficial transfer effects (this was done by asking randomly selected monolingual Italian (*n* = 12) and German (*n* = 12) speakers to rate Latin words and to report if they could deduce their meaning (and, if so, how). The final list was composed based on these ratings). Furthermore, nouns were selected only if they belonged to the first or second Latin declension and showed transparent gender assignment (feminine nouns ending in *-a* and masculine in *-us*) and only if they were plausibly reversible, so that they could be both the agent and the patient of the verb [89]. The main criterion for choosing the adjectives was that the adjective in the corresponding Italian translations would occur postnominally.

Nouns, verbs and adjectives were combined to form sentences aimed at testing the very initial L3/L*n* interlanguage stages. Case morphology and adjective position were selected as linguistic domains of interest (see Section 1.3). Case morphology consisted of *-us* (masculine) and *-a* (feminine) suffixes to mark singular nouns in the nominative position, which correspond to *-um* (masculine) and *-am* (feminine) suffixes in the accusative form (see (4) and (5) exemplar sentences below). Adjectives in these sentences have been assigned the same case as the noun in the second noun phrase (NP) (following Latin rules—1.3). This was carried out to keep sentence length in the EEG experiment constant (i.e., four-word sentences). 
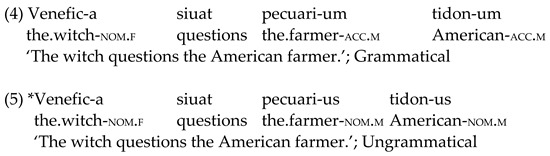


Adjectives were placed either before (ungrammatical) or after (grammatical) the noun in the second NP (see (6) and (7) below). Once again, adjectives have been assigned the case form corresponding to the second NP (see Table 2). 
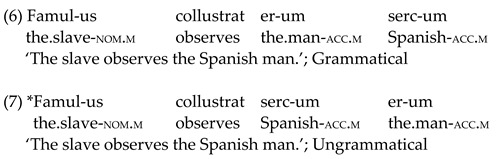


In order to neutralize any other possible confounding factors when creating the sentences for both the training and the ERP phases (see Table 2 below with examples for all experimental conditions), all words within their own category (nouns, verbs and adjectives) have been used the same number of times. Furthermore, word selection was randomized. For nouns, it was ensured that there was the same amount of masculine-only, feminine-only, and mixed-gender sentences. Finally, the study employed visual (as opposed to auditory) training and presentation of stimuli, aimed at removing possible phonological bias (since Latin phonotactics resembles more Italian than German ones). 

### 2.4. Experimental Procedure

#### 2.4.1. Mini-Latin

A pre-training phase, consisting of two vocabulary learning phases (VLP) divided by a consolidation period of 2–10 days, familiarized participants with the nouns, adjectives and verbs of Mini-Latin prior to the EEG data collection. This was carried out to allow the participants to consolidate their newly acquired lexical knowledge, as advocated in [33]. The VLP consisted of four training blocks per session. In each block, participants were exposed to all 28 lexical items of Mini-Latin. Each item was presented in its nominative singular form accompanied by a cartoon picture describing it (e.g., a picture of a bride when presenting the word *nupta*) (see Figure 1). Crucially, neither an Italian nor a German translation was given, in order to avoid any possible language priming effects. Participants were instructed to memorize the item and click a button on the button box when they were ready to move on to the next item. After each block, participants could take a break or directly start with the next one. Every lexical item was presented in capitals (i.e., each letter was capitalized) to avoid possible facilitation and/or priming effects from German (where all nouns are capitalized) (see examples in Table 2). This was not only the case for the VLP, but for the entire Latin experiment (i.e., vocabulary learning phase, grammatical learning phase and grammatical judgment task). Once the four blocks were completed (i.e., participants were exposed four times to each lexical item), participants were tested on their newly acquired vocabulary knowledge. This was assessed via a word-matching task. If they scored above 80% in this task (equivalent to at least 22 correct responses out of 28 total), the training phase was terminated. Otherwise, they were re-exposed to the VLP. Failure to reach the 80% accuracy threshold on a second attempt of the test resulted in the exclusion of the participant from the study. After the consolidation break of 2–10 days, participants returned for the EEG session where they were exposed a second time to the VLP (same process as described above). In total, one participant (2.01%) was excluded from this study during this phase.

In the grammar learning phase (GLP), participants were exposed for the first time (i.e., only in the second session) to the Mini-Latin grammar via a presentation of randomized picture-sentence combinations as in the example sentence below (8) and Figure 2. 



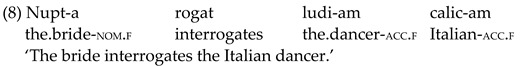



Thus, this phase introduced both nominative and accusative case morphology (critical item *ludiam* in (8)) and adjective position (*calicam*). Exposure to case morphology and adjectival position was achieved through the presentation of 128 meaningful sentences in Mini-Latin without explicit teaching/explanations. These sentences resulted from the crossing of the 28 lexical items of Mini-Latin. The sentences appeared together with 4 series of pictures, where each series was composed of 4 figures. Only one series of pictures depicted the correct lexical order of the sentence. The other 3 were composed so that the resulting picture–word pair did not match: once in the subject position, once in the object position and once in the adjective position (see Figure 2). This was carried out to increase the task’s difficulty, to keep the participants motivated and to avoid facilitating the detection of grammar patterns. Participants were instructed to click on the series of pictures they thought depicted the presented sentence. The training session lasted around 45–60 min. 

At the end of the training, participants were tested on their command of Mini-Latin through a multiple-choice sentence–picture matching task. Upon presentation of a series of 4 pictures, as explained above, participants were asked to select the one sentence that best represented the figure among 5 alternatives (9–13): 

Grammatical sentence 
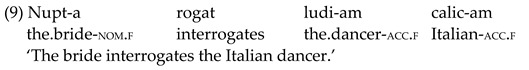


A sentence containing a violation in the case declination 
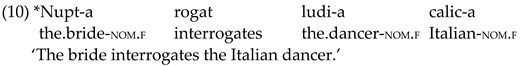


A sentence containing a violation in the adjective–noun order 
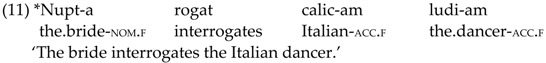


A grammatical sentence containing a lexical violation at the level of the noun 
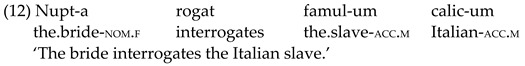


A grammatical sentence containing a lexical violation at the level of the adjective 
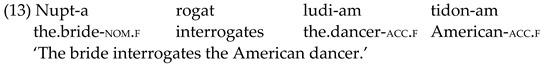


The test consisted of a total of 24 items, in which knowledge of all possible grammatical rules of Mini-Latin was assessed. Test sentences (as in 9–13) were randomized. If participants scored above 80% in this task (equivalent to at least 22 correct responses out of 28 total), they moved onto the experimental phase where EEG was recorded. Otherwise, they were re-exposed to the GLP. Similar to the VLP, failure to reach 80% accuracy a second time resulted in the exclusion of the participant from the study. In total, 4 participants (8.33%) were excluded from the study during this phase.

#### 2.4.2. EEG/ERP

The next phase of the second session was the main experiment with EEG recording. Scalp EEG activity was continuously recorded while participants read the sentences on a computer screen. The rapid serial visual presentation (RSVP) method was employed, i.e., sentences were presented word by word, with an inter-stimulus interval of 250 ms. As soon as a sentence terminated, a black page with a happy and a sad face appeared on the screen indicating that a grammaticality judgment was required. Participants were instructed to press a button on a button box using their left (for happy face—correct sentence) and right (sad face—wrong sentence) index fingers. While response times (RTs) were recorded as an extra measure of behavioral performance, there was no time limit to make the decision. 

The ERP experiment was composed of 168 critical sentences, resulting from crossing the lexicon from Mini-Latin. To form these sentences, lexical items were crossed as randomly as possible, so that all items were repeated the same number of times within their lexical category (nouns, verbs, and adjectives). Four sentence versions (i.e., experimental conditions) were created: grammatical, case violation, adjective violation and double violations (although the latter condition was not analyzed). Violations were either at the level of the noun or at the level of the adjective. Importantly, this was always in the third position in the sentence. Thus, grammatical sentences served as the baseline condition for both case and adjective/noun word order violations. Crucially, this was the first time that participants were consistently (outside from the short test at the end of the GLP) exposed to Mini-Latin grammar violations, i.e., to ungrammaticality to the target properties. This was carried out to better appreciate and capture the degree to which the brain responds to anomalies in a newly acquired (mini) language and, therefore, to elicit representational transfer effects. To prevent any potentially confounding sentence wrap-up effects, a pseudo-randomized 500–1000 ms jittered blank page was set before the grammatical judgment page. This alternative path was followed to avoid the introduction of any type of further noise/distraction in the sentences (see [90] for a review on the falsity of the so-called wrap-up belief/dogma, at least for this type of data). The 168 sentences (42 for each category described above) were divided into four equally long randomized blocks. Participants took breaks between the blocks and were encouraged to rest their eyes and take as long as they needed before continuing with the experiment. 

### 2.5. Adjective and Case Assignment Task

Upon completion of the ERP experiment, participants performed a final task, where their knowledge of morphological case-marking and adjectival position in German (and also adjectival position in Italian for the HSs) was assessed. This was carried out to ensure that these grammatical properties in the participants’ linguistic representations matched the intended values. In each of the three situations, ten grammatical and ten ungrammatical sentences (for a total of 60 items) were presented. In the task, sentences were shown in isolation on the center of the screen, and participants were instructed to press the green button on the button box if they thought the sentence was grammatical, and the red button if otherwise. A short set of three trials familiarized participants with the procedure before every block (representing the three different situations). Accuracy and response times (RTs) were collected for each response. There were no time restrictions in this task and participants were encouraged to favor accuracy over speed. As expected, results showed full alignment across the board with the expected grammatical properties in both German (case morphology and adjective position) and Italian (adjective position—only for the HSs).

### 2.6. Behavioral Data Analysis 

Raw accuracy scores and RTs for the three conditions (grammatical, case and adjective violations) were subjected to statistical analysis. For the RTs analysis only, all data points below 200 ms and/or inaccurate (14.1% of all trials) were excluded from the analysis. Generalized linear models (binomial family) were employed to analyze the accuracy data. Generalized linear mixed models were performed for RTs. 

### 2.7. EEG Recording and Analysis

The EEG signal activity was continuously recorded at the level of the scalp with 32 active electrodes (ActiCap, Brain Products, Inc., Gilching, Germany) fitted in elastic caps following a slightly modified 10–20 system (where 19 electrodes follow the traditional 10–20 system and the rest are placed within the best possible equidistant selection of 10%-positions to increase the topographical distribution). AFz served as the ground electrode, whereas FCz as the online reference. Offline re-referencing was performed to the average of the mastoid (TP9/10) electrodes. Horizontal and vertical eye movements (blinks) were monitored by means of the fronto-polar Fp1 and Fp2 electrodes located just above the eyebrows. A LiveAmp amplifier (Brain Products, Inc., Gilching, Germany) was employed to amplify the signal. Data were recorded with an online filter of 0.01–200 Hz and digitized continuously at a 1 kHz sampling rate. EEG data preprocessing was performed on Brain Vision Analyzer 2.1 (Brain Products, Inc.). All trials were used for the EEG analysis, regardless of accuracy in the grammaticality judgment task (GJT) [33,71,91,92]. First, data were filtered with a band-pass filter of 0.1–30 Hz. Independent component analysis (ICA) was carried out on the whole dataset to take care of eye movements and blinks (512 steps, infomax gradient restricted). After re-referencing to the mastoids, the continuous EEG was segmented into 1500 ms-long epochs with reference to the critical word. Epochs started with a 300 ms pre-stimulus baseline and ended 1200 ms post-stimulus onset. Artifact rejection was carried out semi-manually and then manually inspected (drift, excessive muscle artifact, blinks, blocking, and other artifacts were monitored) and resulted in the exclusion of 0.76% of the total amount of trials. The remaining epochs were first baseline corrected (300 ms relative to the pre-stimulus baseline), then averaged per condition within each participant and finally combined within each group into the grand averages of interest (case violation and adjectival position violation). 

Based on predictions of the L3/L*n* models and findings from previous literature on multilingual language acquisition and processing, the following time windows of interest were isolated by looking at mean amplitudes: 200–500 ms, where the N200/P300 and the N400/left anterior negativity (LAN) components tend to occur [68,93,94] and the 300–600 ms and 400–900 ms, largely corresponding to the typical time window of the P600 [64,68]. The analyses were performed on subsets of electrodes divided into nine regions of interest (ROIs; left anterior: F3/7, FC1/5; right anterior: F4/8, FC2/6; left medial: CP1/5; right medial: CP2/6; left posterior: P1/3/7; right posterior: P2/4/8; midline anterior: Fz; midline medial: Cz; midline posterior: Pz; see, e.g., [33,95] for similar setups). Mean amplitudes for electrodes within these regions were used as dependent variables for the computation of factorial repeated-measures ANOVAs for each time window of interest. Lateral and midline electrode analyses were carried out separately [33,96,97]. ANOVAs included *Condition* (grammatical, case or adjective violation), *Hemisphere* (right, left) and *Caudality* (anterior, medial, posterior) as predictors for lateral electrodes, whereas for the midline the models only included *Condition* and *Caudality*. Huynh–Feldt epsilon estimate corrections were employed in those cases where the assumption of sphericity was violated (see Mauchly’s test) for either a main effect or interaction. This type of correction, while less conservative than Greenhouse–Geisser estimates, is particularly appropriate for those data that inherently have larger between- and within-subject variability [33]. One-way repeated measures ANOVAs were run as follow-up analysis of significant interactions. Analyses of behavioral and ERP data were conducted in R [98] using the following packages: *lme4* for linear mixed models [99], *EMMs* for the post hoc comparisons of the generalized linear mixed models results [100] and *ez* for repeated-measures ANOVAs [101]. 

## 3. Results

### 3.1. Behavioral Data

Accuracy and response times were collected for each condition (grammaticality, case and adjective violations) for both groups. Table 3 provides an overview of descriptive statistics. 

The results for accuracy show a main effect of group (*E* = −1.04, *z* = −8.78, *p* < 0.001), which shows that the subjects from the German L2 group were overall more accurate than the HSs. Further analyses of contrasts show that for both case violation (*E* = 0.79, *z* = 4.45, *p* < 0.001) and adjective violation (*E* = 1.04, *z* = 8.78, *p* < 0.001), the German L2 group performed better than the HSs. For the RT data, we fit generalized linear mixed models with an inversed gaussian family. We began by fitting a maximal model that included condition and group as fixed factors (main effects and interaction), as well as random intercepts for participants and items (Latin words) and random slopes for condition within both. We proceeded to decorrelate random slopes and intercepts and to remove random slopes accounting for the least amount of variance until a model achieved convergence. The final and best fit model included random intercepts for participants and items, but random slopes of condition only for participant. There were no main effects and also no interactions between group and condition of interest (all *p*s > 0.1). In sum, the behavioral data indicate that the German L2 group tends to perform better than the HSs group. Behavioral data have been presented as a form of completeness. However, since the experiment was designed in the first place to obtain maximal explainability at the brain level, from now on moving forward only the neurophysiological data are discussed.

### 3.2. ERP Data

Only main effects or interactions with the *Condition* factor are reported here (i.e., main effects/interactions involving exclusively *Caudality* and *Hemisphere* are excluded) (see Figure 3, Figure 4, Figure 5 and Figure 6 for representative ERP and topographical representations for the two groups). The full output of the omnibus ANOVA for each time window in both groups (HSs and German L2 learners) and conditions (case morphology and adjective position violations against the baselines) can be found in the Appendix A.

#### 3.2.1. Heritage Speakers

##### Case Morphology


*200–500 ms*


A very strong main effect of *Condition* (*F*(1,14) = 18.7, *p* < 0.001, *η^2^* = 0.1) in the lateral electrodes was observed in the 200–500 ms time window for the comparison between case morphology and baseline conditions in the HSs. The analysis of the midline electrodes yielded similar results, with a main effect of *Condition* (*F*(1,14) = 13.5, *p* = 0.003, *η^2^* = 0.07). In both cases, interactions with *Hemisphere* and *Caudality* were not significant, meaning the increases in negative voltage in lateral and midline electrodes in the case violation condition with respect to the baseline were evenly distributed (Figure 3 and Figure 4). 


*300–600 ms*


Once again, the analysis of case morphology against the baseline condition resulted in a trend and a main effect of *Condition* for both lateral (*F*(1,14) = 14.2, *p* = 0.002, *η^2^* = 0.08) and midline (*F*(1,14) = 6.87, *p* = 0.02, *η^2^* = 0.04) electrodes, followed by no interactions with either topographical factors. This means that an increase in voltage for case violations in comparison to the baseline was found across the whole scalp (Figure 3 and Figure 4). 


*400–900 ms*


In contrast to the two previous time windows, no significant effect of *Condition* was found for both lateral (*F*(1,14) = 1.1, *p* = 0.31) and midline (*F*(1,14) = 0.0, *p* = 0.92) electrodes. These findings suggest that case morphology violations did not elicit a different response from the grammatical condition in the 400–900 ms time window. 

##### Adjective Position 


*200–500 ms*


For the time window 200–500 ms and comparison between adjective order violations and baseline in the HSs, no main effects of *Condition* were found for both lateral (*F*(1, 14) = 1.58, *p* = 0.23) and midline (*F*(1,14) = 0.48, *p* = 0.5) electrodes. Further, no interaction effects for both lateral and midline electrodes were found with either of the topographical factors.


*300–600 ms*


Similar to the previous time window, no main effects of *Condition* were found here for both lateral (*F*(1,14) = 1.28, *p* = 0.28) and midline (*F*(1,14) = 0.38, *p* = 0.55) electrodes. Consequently, no interactions with *Hemisphere* and *Caudality* were found for both electrode positions.


*400–900 ms*


As in the previous two time windows, there were no main effects of *Condition* for neither lateral (*F*(1,14) = 1.12, *p* = 0.3) nor midline (*F*(1,14) = 0.28, *p* = 0.6) electrodes, as well as no two-way or three-way interactions with this factor. This indicates that adjective position violations did not elicit a different response from grammatical sentences in the 400–900 ms time window.

#### 3.2.2. German L2 Learners

##### Case Morphology 


*200–500 ms*


No significant main effect of *Condition* nor any two-way and/or three-way interactions with *Hemisphere* and *Caudality* were found in the analysis of the lateral and midline electrodes.


*300–600 ms*


No significant main effects of *Condition* were found in the lateral (*F*(1,26) = 0.56, *p* = 0.46) or midline (*F*(1,26) = 0.63, *p* = 0.43) electrode sites in the 300–600 ms time window. Likewise, there were no significant interactions of *Condition* with either of the topographical factors (*Hemisphere* and *Caudality*).


*400–900 ms*


As in the other two time windows analyzing case morphology violations against the baseline in the German L2 group, no significant main effects in either lateral (*F*(1,26) = 0.09, *p* = 0.76) or midline (*F*(1,26) = 0.27, *p* = 0.6) electrodes were found. Similarly, no significant interactions with *Hemisphere* and *Caudality* were observed.

##### Adjective Position 


*200–500 ms*


In the lateral electrodes, the omnibus ANOVA revealed a main effect of *Condition* (*F*(1,26) = 6.45, *p* = 0.02, *η^2^* = 0.03) for adjective position violations in the 200–500 ms time window. This indicates that scalp voltages were more positive in the adjective violation condition as compared to grammatical sentences. This effect seems to be broadly distributed, as all two-way and three-way interactions with the topographical factors were not significant. A similar significant main effect of *Condition* was found in the analysis of the midline electrodes (*F*(1,26) = 6.05, *p* = 0.02, *η^2^* = 0.03). Additionally, here, the two-way interaction between *Condition* and *Caudality* was not significant (*F*(2,52) = 0.53, *p* = 0.59). This indicates an increase in positive voltage (Figure 5 and Figure 6) evenly distributed along the midline electrodes in adjective violations compared to the baseline.


*300–600 ms*


A main effect of *Condition* (*F*(1,26) = 7.57, *p* = 0.01, *η^2^* = 0.03) was found in the lateral electrodes in the omnibus ANOVA. Since no two-way or three-way interactions with the other factors were significant, this effect indicates an increase in voltage distribution over the whole scalp for adjective violations in comparison to the baseline. Similarly, the analysis of the midline electrodes yielded a main effect of *Condition* (*F*(1,26) = 5.45, *p* = 0.03, *η^2^* = 0.01), but no interaction of *Caudality* (*F*(2,52) = 0.47, *p* = 0.63), indicating that this increase in positive voltage was equally distributed (Figure 5 and Figure 6). 


*400–900 ms*


Two trend effects of *Condition* were observed for the lateral (*F*(1,26) = 4.04, *p* = 0.054, *η^2^* = 0.02) and midline electrodes (*F*(1,26) = 3.47, *p* = 0.07, *η^2^* = 0.02) in this time window for the comparison between adjective violation and baseline voltages. However, no two-way and three-way interactions were found for either lateral or midline electrodes, indicating that the trends for the increase in voltage between adjective violations and baseline conditions were uniformly distributed (Figure 5 and Figure 6).

### 3.3. Summary of ERP Results

Case violations elicited in the HSs group were strongly and broadly negatively distributed main effects in both 200–500 ms and 300–600 ms time windows, which then dissipated in the latest period. No effects were found for the German L2 group across the board. Adjective placement violations, by contrast, produced distributed positivities in the first two time windows in the German L2 group and no effects in the HSs. 

## 4. Discussion

The data presented above do not seem to conform straightforwardly to the expectations of the TMP or the LMP (or any of the current L3/L*n* models for that matter). However, they depict neurophysiological differences between the two groups and the two linguistic domains. In what follows, findings are unpacked in two steps. We first discuss how the findings deviate from our predictions and what this means at the theoretical and methodological levels. We then explore what the data reveal for the very initial stages of interlanguage L3/L*n* transfer and processing. Although the present results do not allow us to adjudicate between the existing L3/L*n* models that we sought to test, we argue that they offer crucial insights into the very earliest stages of multilingual language acquisition.

### 4.1. Revisiting the Initial Predictions

Following the TPM (holistic transfer model), it was posited that Italian would be the source of transfer for the HSs, and that English would be the transfer source for the German L2 learners. Thus, the TPM predicted sensitivity (N400-P600 biphasic response or P600) to the adjectival position violations for the HSs and no sensitivity for either domain for the German L2 group. Alternatively, the LPM/SM (property-by-property transfer models) predicted German transfer for case morphology in both groups (i.e., either N400-P600 biphasic responses or a P600) and sensitivity to adjective violations for the HSs (see Table 1). Similar to what was observed in [33], these ERP predictions were not attested. The early N200/N400 broadly distributed negativity for case was not anticipated by any of the models (at least not in this form). Indeed, despite both groups knowing German where the relevant morphosyntactic features are instantiated (though not exactly in the same form—see Section 1.3) and thus the underlying features were, in principle, available for transfer, neither group showed grammatical transfer effects. The LPM/SM would have predicted some effect from German for case, although it is possible (and might matter for transfer effects) that German and Latin realize these features differently in their surface form (see Section 1.3). In any case, the qualitative nature of the signature observed—early negativity—is not reminiscent of morphosyntactic processing per se. The fact that it also only occurs for the HSs, despite both groups being dominant in German, motivates us to ponder alternative understandings of this effect.

Similarly, the broadly distributed P600 for adjectival position observed for the German L2 group only is not anticipated by either the TPM or the LPM/SM. This is the case because there is no previous source of transfer that could have provided transfer features for this group. While this word order asymmetry is represented in the Latin stimuli themselves and, thus, could have been acquired/learned in the course of the experiment, the question remains why only the German L2 group learned it. Minimally, following the LPM/SM, the Italian HSs should have been quicker in acquiring this domain—facilitation at the property level—and, following the TPM, the HSs should have shown early sensitivity. Thus, the fact that only the German L2 group shows this sensitivity is an equal quandary for both models. 

In summary, no current L3/L*n* model predicted the pattern of findings from the presented data. Importantly, all predictions were posited on the assumption that transfer would have already taken place during testing. However, we maintain that the findings instead seem to indicate that the amount of exposure, even with consolidation between the two lexical learning sessions (as advocated in [33]), was not sufficient in the current design to be tapping the first stage(s) of an emerging L3/L*n* interlanguage. Instead, they might be showing processes of attending to linguistic stimuli at a stage prior to when transfer is likely to occur in an L3/L*n* scenario (as argued in the only other comparable study using EEG [33]). 

### 4.2. Expanding the Findings beyond the L3/Ln Models

No L3/L*n* study so far, to the best of our knowledge, has claimed to show evidence of transfer after such minimal exposure (around 1 h). This might be the case because none of the current models of L3/L*n* acquisition offer precise estimations regarding the timing of transfer selection. On the one hand, (i) the TPM’s cue hierarchy, for example, provides a rubric of timing closely dependent on the overall similarity (typology) between the L3 and the source languages. Moving through the different levels of the TPM’s hierarchy requires both increasing exposure and time. This, in turn, means that there will be inevitably a delay in time in language triads where the L3 does not display a close resemblance to either the L1 or the L2 (at least at the lexical and/or phonological levels). On the other hand, models such as the LPM/SM assume incremental learning as a result of processing, making early unstable representations increasingly stable. Given incremental learning, the LPM would not expect different properties to stabilize at exactly the same time, but to be dependent on factors such as input frequency and complexity. Moreover, this also means that the best (if not the only) way to test the LPM would be to follow developmental L3/L*n* learning over time. Under both models, however, it is not immediately clear that the paradigm of mini grammar learning, at least a non-longitudinal approach as undertaken herein, is optimal to test the predictions of either model independently or indeed against one another. Extending out the learning of Mini-Latin as a longitudinal study would potentially remedy some of the inherent confounds, especially making it able to test the claims of differences in increasing stability of properties between sets of learners. As the method was deployed, it is possible, if not likely, that all that was observed reflects how the participants approached *the doing* of this specific task itself. On the one hand, the HSs seem to be more sensitive to overt morphological marking overall, resulting in greater attentional resources focusing on case anomalies. On the other hand, the German L2 learners were somehow better at intuiting the Latin rule of adjectival ordering. However, none of this is a direct reflection of linguistic transfer effects per se. 

In light of the above, RQ1 *“Can neurophysiological signatures (EEG/ERPs) inform models of multilingual morphosyntactic transfer?”* is complicated to answer. Based on our data, it can be simply said that the present experiment was not ideal to be informative. However, this certainly does not mean that outside of a mini grammar paradigm (or potentially with it in a longitudinal application) EEG/ERPs as a method are not effective for L3/L*n* research. Nevertheless, even the present data are relevant for addressing the spirit of what was asked in RQ2: *“What do data from HSs bring to bear for answering RQ1 as well as understanding how they might differ from L2 learners when acquiring additional languages in adulthood?”*. Below, the second part of this question and the two most salient patterns of results are discussed (N200/N400 for case for the HSs and P600 for adjective position for the German L2 learners).

The ongoing, broadly distributed negativity starting in the 200–500 ms window and stretching over to the 300–600 ms for the HSs in case violations can be interpreted in two different ways. First, this result could signal an N400. Although this component usually tends to be maximally present over centro-parietal regions, it is not uncommon to find it within a more widespread scalp distribution [102,103,104,105]. Studies in language processing and acquisition in monolingual native speakers tend to find P600 effects alone or with a left lateralized early negativity (LAN) [64,65,67]. However, even though mostly found in developmental data of non-native speakers, N400 effects alone or combined with a later P600 have been reported for properties such as case [70,71]. This would be a plausible explanation for the N400 effect, indicating early grammatical transfer from German in property-by-property models. However, this is only plausible if, at the same time, a similar effect had been found in the German L2 learners (see Table 1). Given that this was not the case, how do we interpret what is observed in the HSs? 

Answering this question, which leads to the second interpretation of the early negativity, has two possible explanations. On the one hand, it might indeed be a classic N400, but rather than any interlanguage transfer effects, it highlights task-learning patterns in a broader sense (although this is not the most common view for the N400, but see [106,107]). On the other hand, and possibly more intriguing, the data might indicate a long-stretched N200 component. This is more likely considering that the onset of the early negativity is earlier than one would expect for a classic N400 (see Figure 3 and Figure 4). The N200 component, sometimes reported as “N2-P3 complex” (i.e., coupled with the P300 component), signals three different types of processes depending on the task/condition: (1) a fronto-central (anterior) component related to attention and detection of novelty/mismatch from a perceptual known template, (2) a second fronto-central component related to cognitive control (encompassing response inhibition, response conflict, and error monitoring), and (3) a posterior N200 related to aspects of visual attention [108,109]. Given the nature of the task herein and the topography found, we posit that this is likely an N200 of type (1). Studies employing oddball paradigms or, in general, some tasks where complex novel stimuli are presented in contrast to habituated situations/stimuli find patterns of results consistent with a description of the N2 *novelty* effect as arising from a “deviation from a predominant stimulus category” [110,111,112]. Similarly, in matching tasks, *template-matching* effects are observed, i.e., N200 enhanced responses are detected when the second stimulus in a pair does not match the first [113,114,115]. 

What could this mean in the context of the present experiment? We submit that HSs are directing their attention differentially, thus picking up some type of novelty/mismatch in the Mini-Latin language, specifically where it relates to a surface morphological form pattern mismatch. Recall that by the time of testing, participants were never exposed to ungrammatical structures. The N200 might, therefore, be a sign of allocating of early attentional resources and detection of novelty/mismatch in ungrammatical versus acquired/learned Mini-Latin properties. Since only HSs are displaying this behavior, the question is why only this group. First, since the main linguistic difference between the two groups is the presence of Italian (and Italian grammar) in the HSs (both groups having German as a native language and mid-high English proficiency), and since Italian has a widespread “matching” (declension and inflectional morphology) system at all lexical levels (nouns, adjectives, articles/determiners and pronouns) [116,117], it might follow that knowledge of Italian helps them in tuning and matching the observed patterns in Mini-Latin. However, this is likely not the whole story given that German also has quite rich morphological paradigms. Therefore, if there is an effect of Italian in this general regard, it is likely a cumulative one (having two relatively speaking rich morphological languages) that increases the HSs’ attention to morphological patterns overall. Secondly, the observed differences in novel language learning might arise due to the longer experience the HSs have with being bilinguals in comparison to the German L2 group. There is no verifiable way to know with the present data which position better explains the pattern of results, but there are reasons to highlight this as a valuable (and viable) inquiry for future research. 

The P600 in adjective/noun position violations also cannot be straightforwardly explained as a sign of representational transfer because it was obtained only for the German L2 learners whose previously acquired language does not provide the grammatical features/structure for transfer. If anything, both kinds of models (TPM—holistic vs. LPM/SM—selective transfer) would predict the opposite effects, i.e., transfer from Italian and, therefore, for the HS group only. The only possible explanation is that the German L2 learners have managed to learn in such a short amount of time the adjective/noun rules of the mini language (see above for a similar interpretation for the N200/N400 result). In this case, the P600 highlights signs of repair/integration of the pattern being broken (adjective–noun position violation with respect to the baseline condition). It might be the case that the German L2 learners were able to focus more on this because the word order is odd in both German and English—the languages they knew—therefore making it stand out more in the Latin input. Alternatively, because for this property the HSs know a language that allows one or the other order, the pattern of N–A was not salient (yet) as it was for the German L2 learners. Note that the P600 is not an index of linguistic processing per se, but rather reliably appears when (linguistic) repair is needed. Thus, the P600 observed here is compatible both with learning of the rule as a linguistic one—where its violation induces morphosyntactic repair—or as a pattern rule that is not linguistic per se but induced a similar type of repair. We are more inclined toward the former possibility considering the reduced exposure and the controlled grammar in the experimental conditions. Since this could not come from linguistic transfer, as discussed, it is much more likely to be reflecting pattern matching anomaly detection. In other words, we believe that the early negativity and the P600 reflect processes of how the two groups approached the experimental conditions/paradigm themselves more than linguistic-specific processing per se.

If we are on the right track, it seems to be the case that such a short training exposure is insufficient to tap into what we originally hoped to, that is, processes of multilingual interlanguage transfer. Rather, the observed results seem to reflect less linguistically and more learning-wise approaches to accomplishing the task at hand.

This study is not without limitations. First, while the relatively small sample was determined to be sufficiently powered for the analyses used herein, larger sample sizes will be required in order to run more complex analytical approaches (e.g., trial-by-trial analysis instead of ANOVAs). Second, it has been criticized within the L3/L*n* field that most of the studies investigate similar language combinations (mostly combining English with either Germanic and/or Romance languages). This has led some scholars to argue that findings might be skewed towards one or the other model. While the usage of a Mini-Latin language is a relative novelty within the field, we acknowledge that Latin might prime Italian transfer over German due to their overall similar linguistic structure (but, to be fair, that is the very contention of at least one of the models, the TPM). To address this, however, we carefully selected non-cognate words between the languages, among other things (see Section 2.3). However, we of course agree that widening the spectrum of linguistic variety pairings tested in the field will increase validity, power and knowledge within multilingual language acquisition/processing. Finally, although we tried to select participants who did not speak any other language beyond German and English for the German L2 group and German-Italian and English for the HSs, we recognize that some participants have had other languages in school other than the ones investigated herein. In a world where multilingualism is the norm, avoiding such potential confounds is difficult and potentially not warranted. Alternatively, it must be acknowledged, documented, considered in the analysis and, where possible leveraged. In the present case, participant selection permitted other languages studied when they reported to not actively use them. Future studies should try to increase sample size to allow trial-by-trial analyses (see above) to be able to regress sophisticated linguistic background information to brain data, thus increasing the granularity and precision of both the research questions and the related statistical approaches [118].

## 5. Conclusions

The findings herein do not support any of the models of L3/L*n* language acquisition and transfer, but they do not speak against them in any meaningful way either. In fact, the very nature of the current methodological paradigm, at least how we and others have thus far applied the mini grammar learning method in L3/L*n*, seems unable to capture the effects we were probing for. This might be the byproduct of limiting experimental choices and/or an artifact of study limitations. 

Thus, if future studies aim at pursuing this route, they must increase the exposure time and introduce multiple stages of consolidation for the (mini) language, making them, ideally, longitudinal. Increasing exposure to several consolidation periods will make the mini grammar paradigm more ecologically valid, i.e., more similar to real language learning, and permit time for transfer effects to kick in. Increasing the number of linguistic properties under investigation (taking into consideration different structural complexity and how they relate to the L1 and L2) and examining development over time will increase the chances of meaningfully testing models against one another. Such an approach can capture both an ab initio L3/L*n* interlanguage grammar to determine if transfer is then complete as argued by the TPM, as well as later developmental L3/L*n* as predicted by models such as the LPM/SM (i.e., whether in different groups specific properties are stabilized sooner or later during the linguistic multilingual development). Electrophysiological measures should be assessed in more than one session in order to follow the development and capture possible neural changes at all levels of language acquisition and processing (sensory, higher-order processing, interlanguage grammar transfer). 

## Figures and Tables

**Figure 1 brainsci-12-00669-f001:**
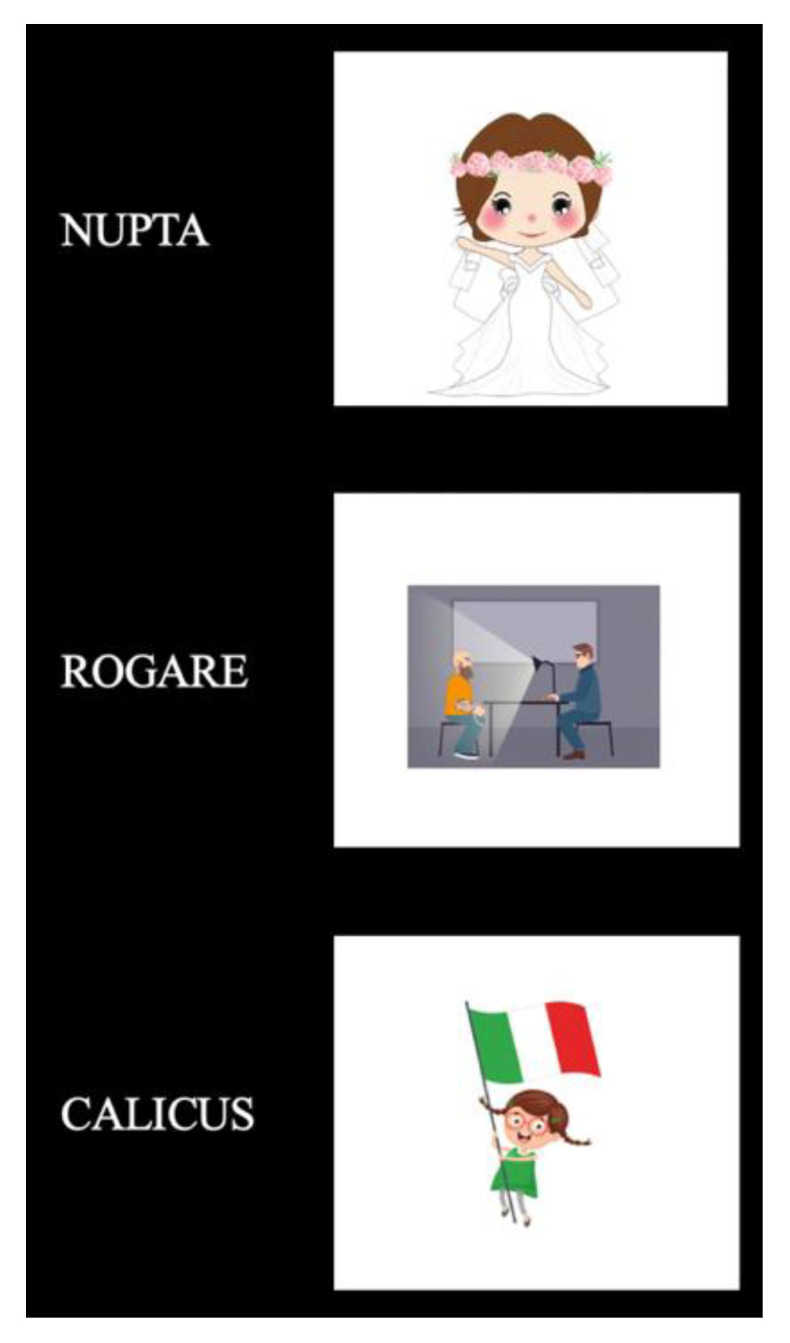
Three distinct examples (noun (**top**); verb (**center**); adjective (**bottom**)) as presented in the vocabulary learning phase (VLP).

**Figure 2 brainsci-12-00669-f002:**
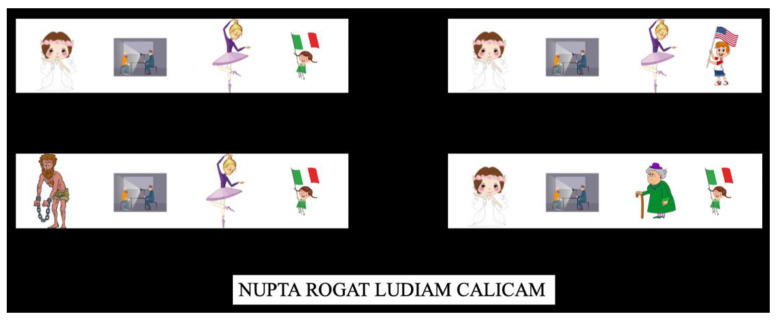
Example of one trial of the grammar learning phase (GLP). Participants were required to select with the mouse the series of figures that represented the correct lexical order of the sentence. (**Top left**) Correct lexical order of the sentence. (**Bottom left**) Wrong picture–lexical match in the subject position. (**Top right**) Wrong picture–lexical match in the adjective position. (**Bottom right**) Wrong picture–lexical match in the object position.

**Figure 3 brainsci-12-00669-f003:**
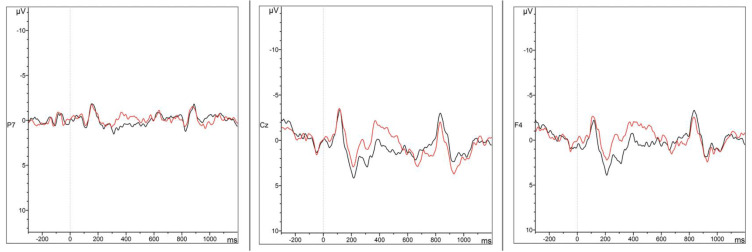
Grand Average ERP waveforms for the grammatical (**black**) and case morphology (**red**) conditions in Mini-Latin at electrodes P7, Cz and F4.

**Figure 4 brainsci-12-00669-f004:**
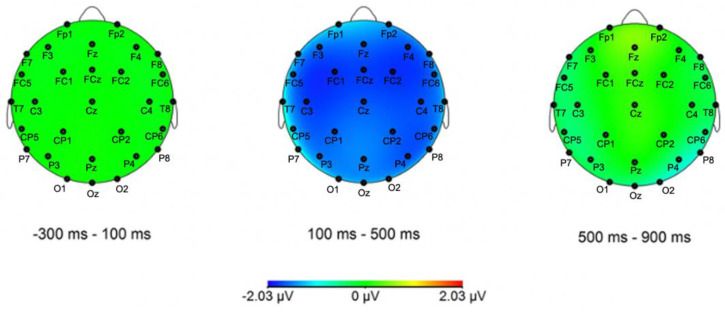
Topographical distribution maps of the difference wave for the case-grammatical condition contrast for the HSs group in the Mini-Latin experiment. All electrodes, apart from TP9/TP10 (offline references) and FT9/FT10 (falling outside of the +/−90 degrees plane due to sphericity constraints), are labeled in the topographic maps.

**Figure 5 brainsci-12-00669-f005:**
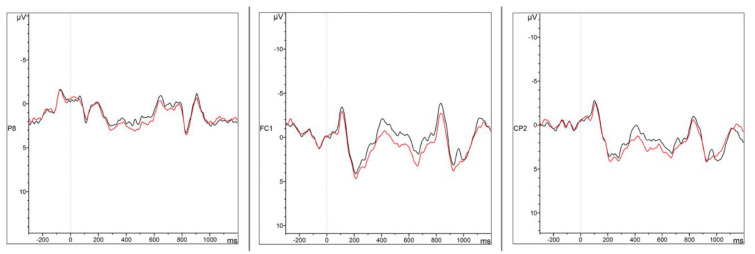
Grand Average ERP waveforms for the grammatical (**black**) and adjective position (**red**) conditions in Mini-Latin at electrodes P8, FC1 and CP2.

**Figure 6 brainsci-12-00669-f006:**
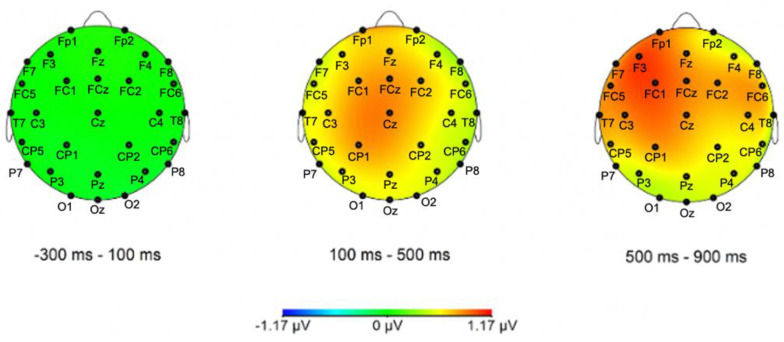
Topographical distribution maps of the difference wave for the adjective position-grammatical condition contrast for the German L2 group in the Mini-Latin experiment. All electrodes, apart from TP9/TP10 (offline references) and FT9/FT10 (falling outside of the +/−90 degrees plane due to sphericity constraints), are labeled in the topographic maps.

**Table 1 brainsci-12-00669-t001:** ERP predictions for the processing of case morphology and adjectival position violations in Mini-Latin by adult HSs of Italian in Germany (2 L1s—Italian and German) and L1 German–L2 English late L2ers, derived from the TPM, LPM and SM (see also [31,33]).

			*Case* *Morphology*			*Adjective* *Position*	
Language combination		TPM		LPM/SM	TPM		LPM/SM
L1 Italian—L1 German (L2 English)—L3 Mini-Latin		No effect		(N400)-P600 *	(N400)-P600		(N400)-P600
L1 German—L2 English		No effect		(N400)-P600 *	No effect		No effect
L3 Mini-Latin							

* Although the underlying syntactic features are the same, if the surface exponent is taken to matter—case being expressed mostly in determiners/adjectives in German, whereas it also agrees on the noun in Latin—it could be that these models would expect no effect.

**Table 2 brainsci-12-00669-t002:** The four experimental conditions for the Mini-Latin experiment (see glosses and translations above).

Test Sentence	Experimental Condition
VENEFICA SIUAT PECUARIUM TIDONUM	Grammatical
VENEFICA SIUAT PECUARIUS TIDONUS	Case violation
VENEFICA SIUAT TIDONUM PECUARIUM	Adjective violation
VENEFICA SIUAT TIDONUS PECUARIUS	Double violation

**Table 3 brainsci-12-00669-t003:** Mean accuracy (%) and reaction times (ms) with standard deviations for the three analyzed conditions in both groups.

		Accuracy (%)		Reaction Times (ms)	
Group	Condition	Mean	SD	Mean	SD
German L2 group	Grammatical	95.2	21.5	768	798
	Case violation	94.4	23	618	984
	Adjective violation	85.5	35.3	861	1288
Heritage Speakers	Grammatical	94.9	22	534	1098
	Case violation	88.4	32	454	594
	Adjective violation	67.5	46.9	642	973

## Data Availability

Publicly archived datasets will be created in accordance with the Data “MDPI Research Data Policies” if this study is accepted for publication in *Brain Sciences*.

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
