# Peer review of "Testing Potential Transfer Effects in Heritage and Adult L2 Bilinguals Acquiring a Mini Grammar as an Additional Language: An ERP Approach"

_brainsci, 2022, doi:10.3390/brainsci12050669_

Round 1

Reviewer 1 Report

See the attachment.

Author Response

Dear Reviewer #1,

Thank you very much for your valuable comments. We have found them extremely helpful and we believe they have led us to substantially improve the quality of the paper, including rearranging the introduction section, increasing the precision of detail of the methods section, while at the same time cutting down on the size and improving the language. We have now amended the manuscript and highlighted in red the changes based on the reviewer’s comments (from you and from other reviewers).

Please find the revised manuscript attached.

With best regards,

The authors

Reviewer 2 Report

Main message

The paper reported a study examining language acquisition in heritage speakers and L2 learners using the EEG. The paper reported differences in heritage speakers and bilinguals in terms of the ERPs observed where N200/N400 deflection for the HSs in Case morphology and a P600 effect for the German L2 group in adjectival position. The findings did not align with the hypotheses but indicated differences between heritage speakers and L2 learners in further language acquisition.

General judgment comments

Language acquisition is an important area of study and the mechanisms underlying language acquisition are still not well understood. Hence, the paper can contribute novel information using the EEG to this field. However, some major revisions are required in order to further strengthen the paper.

Major issues

1) Introduction

  • The flow of the introduction can be confusing at times. Although Section 1.2 is labelled “Online Brain Methods and Mini Grammar Paradigm”, the mini grammar paradigm was not explained in detail in this section. Greater clarity of ideas could be achieved by providing a brief description of the mini grammar paradigm first before explaining how it was employed specifically in the context of the present study.
  • The introduction is currently too long and can be more succinct. Much of the information can be summarized in a more concise manner. Section 1.4 on the comparison between Italian and German is also somewhat excessive and can be summarized.

2) Methods

  • Considering that the study was looking at mainly L3 acquisition, did the authors exclude individuals who had already acquired L3? Or is there information regarding the number of languages that participants had already acquired? For example, did HSs recruited in the study only speak Italian and German?
  • The authors mentioned that a total of 5 participants were excluded from the study due to the failure to achieve 80% accuracy for either the vocabulary or grammar learning task. The authors should report the final sample included for the data analysis and indicate the final numbers of HSs and L2 learners.
  • The authors mentioned that there were nine ROIs – how were these ROIs selected? The authors also did not mention ROIs that were examined in previous studies as well as the results in their literature review.

3) Discussion

  • The authors did not discuss any limitations related to their study. Firstly, the final sample size of the study appears to be rather small. Secondly, the languages investigated in the present study share the same writing system and it should be pointed out that L3 acquisition may be potentially different across languages of different writing systems and origins. Thirdly, in terms of HSs - considering that the HSs were Italians living in Germany, did they study German in school as well? The amount of time speaking/learning a particular language may influence the transfer between languages.

Minor issues

1) The manuscript is too long in general; particularly, the introduction and discussion are somewhat excessive and can be made to be more concise.

2) Line 105: The first instance of heritage speakers should be accompanied with its abbreviation – i.e. heritage speakers (HSs).

3) It would be good to proofread the paper for language errors. Some of the sentence structures also made the meaning of the sentences difficult to comprehend.

Author Response

Dear Reviewer #2,

Thank you very much for your valuable comments. We have found them extremely helpful and we believe they have led us to substantially improve the quality of the paper, including rearranging the introduction section, increasing the precision of detail of the methods section, while at the same time cutting down on the size and improving the language. We have now amended the manuscript and highlighted in red the changes based on the reviewer’s comments (from you and from other reviewers).

Please find the revised manuscript attached.

With best regards,

The authors

Reviewer 3 Report

This paper presents interesting findings regarding models of language learning in regards to transfer and timing. The study uses EEG/ERPs to examine different theories by discussing the results of and experiment that brought together two types of bilinguals exposed to a novel L3/Ln (Mini-Latin).
Participants were tested via EEG/ERPs to investigate underlying grammar processing and morphosyntactic representations. Neurophysiological findings showcase a N200/N400 deflection for the HSs in Case 20
morphology and a P600 effect for the German L2 group in adjectival position. As the authors note, no current L3/Ln theories predict the observed pattern of results, which, as the authors mention, brings to mind questions surrounding the appropriateness of using this method for adjudicating between the existing models rather than anything specific about the models themselves. The resulting patterns illuminate differences in how HSs and L2 learners might approach the very initial stages of additional language learning.

Given that I found the paper clear and compelling I am recommending publication in nearly it's current form. However in line 50 the word "alludes" should be "implies" and in line 67 it is somewhat unclear what "system" you are referring to. 

Author Response

Dear Reviewer #3,

Thank you very much for your very positive feedback. We have now amended the manuscript and highlighted in red the changes based on the reviewer’s comments (from you and from other reviewers).

Please find the revised manuscript attached.

With best regards,

The authors

Round 2

Reviewer 1 Report

The EEG methodology (in particular the way to avoid that the stimulus-related components contaminate the event-related ones) must be further refined.

Author Response

Dear Reviewer #1,

Thank you very much for your valuable comment. In what follows, we hope that we can better explain the reasons that led us to perform the neurophysiological steps as we did.

Please find the revised manuscript attached (style/spelling check performed).

With best regards,

The authors

Reviewer 2 Report

interesting article

Author Response

Dear Reviewer #2,

Thank you very much for finding our study interesting. We have performed a style/spelling check throughout the manuscript. We have now amended the manuscript and marked the changes based on the reviewer’s comments (see track changes).

Please find the revised manuscript attached.

With best regards,

The authors
